# Propionic Acid and Fasudil as Treatment against Rotenone Toxicity in an In Vitro Model of Parkinson’s Disease

**DOI:** 10.3390/molecules25112502

**Published:** 2020-05-28

**Authors:** Friederike Ostendorf, Judith Metzdorf, Ralf Gold, Aiden Haghikia, Lars Tönges

**Affiliations:** 1Department of Neurology, St. Josef-Hospital, Ruhr-University Bochum, 44791 Bochum, Germany; friederike.ostendorf@rub.de (F.O.); judith.metzdorf@rub.de (J.M.); ralf.gold@rub.de (R.G.); aiden.haghikia@rub.de (A.H.); 2Neurodegeneration Research, Centre for Protein Diagnostics (ProDi), Ruhr University, 44801 Bochum, Germany

**Keywords:** neurodegeneration, Parkinson’s disease, rho kinase, rotenone, short chain fatty acids

## Abstract

Parkinson’s disease (PD) is a multifactorial neurodegenerative disease. In recent years, several studies demonstrated that the gastroenteric system and intestinal microbiome influence central nervous system function. The pathological mechanisms triggered thereby change neuronal function in neurodegenerative diseases including dopaminergic neurons in Parkinson´s disease. In this study, we employed a model system for PD of cultured primary mesencephalic cells and used the pesticide rotenone to model dopaminergic cell damage. We examined neuroprotective effects of the Rho kinase inhibitor Fasudil and the short chain fatty acid (SCFA) propionic acid on primary neurons in cell morphological assays, cell survival, gene and protein expression. Fasudil application resulted in significantly enhanced neuritic outgrowth and increased cell survival of dopaminergic cells. The application of propionic acid primarily promoted cell survival of dopaminergic cells against rotenone toxicity and increased neurite outgrowth to a moderate extent. Interestingly, Fasudil augmented gene expression of synaptophysin whereas gene expression levels of tyrosine hydroxylase (TH) were substantially increased by propionic acid. Concerning protein expression propionic acid treatment increased STAT3 levels but did not lead to an increased phosphorylation indicative of pathway activation. Our findings indicate that both Fasudil and propionic acid treatment show beneficial potential in rotenone-lesioned primary mesencephalic cells.

## 1. Introduction

Parkinson’s disease (PD) is the second most common neurodegenerative disease [1]. The clinical symptoms, such as resting tremor, bradykinesia, rigidity, and postural instability, result to a large extent from the loss of dopaminergic neurons in the substantia nigra. Another hallmark is the presence of Lewy bodies and Lewy neurites, which mostly consist of misfolded alpha Synuclein (aSyn). Interestingly, non-motor symptoms like hyposmia or depression often manifest already before motor symptoms arise [2,3]. Gastrointestinal dysfunctions such as constipation are other common manifestations [4]. While the etiology of PD is not yet completely understood, it is suggested that both genetic and environmental factors contribute to the pathogenesis [5].

Since the publication of the Braak hypothesis of aSyn propagation from gut to brain [6], the gut–brain-axis has become an important focus of PD research. Alterations of the gut and its microbiome emerged as unique factors in PD pathogenesis. Clinical studies show alteration in the microbiome in fecal samples of PD patients compared to healthy controls [7]. Especially, the role of short chain fatty acids (SCFA) such as acetic acid, propionic acid, and butyric acid as metabolites of enteric bacteria have recently been studied in human PD and in various animal models with heterogeneous descriptive results [8,9,10,11]. SCFA are produced by anaerobic bacteria like *Fusobacterium* through the fermentation of dietary carbohydrates such as dietary fibers [12,13]. The naturally occurring propionic acid is used as food preservative, due to its fungicide and bactericide potential [14]. Besides direct effects in the gut, SCFA are detectable in the peripheral blood and are suggested to directly influence the blood–brain barrier [15]. However, direct effects of SCFA on cultured mesencephalic neurons under stress conditions and thereby activated mechanisms have only been incompletely studied.

To model the degeneration of dopaminergic neurons in in vitro experiments different toxin-based paradigms are available. An often-used toxin is 1-methyl-4-phenylpyridinum (MPP+), which is selectively taken up by dopaminergic cells, where it concentrates in mitochondria and leads to inhibition of the electron transfer chain [16]. MPP+ is a toxic metabolite of the compound 1-methyl-4-phenyl-1,2,3,6-tetrahydropyridine (MPTP), which was first described to cause parkinsonism in 1983 by Langston et al. [17]. Apart from MPP+, the pesticide rotenone confers its toxic effect by inhibition of complex I and thereby causes oxidative stress to dopaminergic cells [18]. In contrast to MPP+, rotenone has been shown to facilitate aggregation of aSyn [19] and is a known environmental risk factor for PD [20,21].

In our experiments we cultured primary mesencephalic neurons of rat embryos under stress conditions with rotenone to examine lesion mechanisms to dopaminergic neurons. In order to evaluate novel therapeutic strategies in vitro, we took advantage of the rotenone toxicity model and applied two different strategies. Propionic acid was used as a potential regenerative treatment, as SCFAs have already been discussed for their influence on immune reactions in the gut [9] and have been demonstrated to elicit a direct effect on neurons after passing the blood–brain barrier [22]. Propionic acid in particular is mainly metabolized in the liver but around 10% enters the bloodstream and can be detected in the peripheral blood [12]). Because of its potential to cross the blood–brain barrier [23], we examined the direct effect of propionic acid on neuronal cells of the central nervous system. As a second treatment we used 5-(1,4-Diazepane-1-sulfonyl) isoquinoline (Fasudil), a pharmacological inhibitor of the Rho-associated protein kinase (ROCK), which is an effector of small GTPase RhoA that regulates actin cytoskeleton organization. Inhibition of ROCK has shown to mediate neuritic outgrowth, axonal regeneration, neuronal cell growth, and neuroprotection due to reduced induction of apoptosis [24]. It is known to be able to protect and facilitate regeneration of primary dopaminergic neurons after treatment with the toxin MPP+ [25]. In order to evaluate possible neurite growth promoting effects with propionic acids, Fasudil, and their combination, we applied these substances to the rotenone toxicity model in vitro. Thereby, we sought to analyze their potential to foster neurite outgrowth and the involved pathways in order to prepare future therapeutic studies for disease modification in PD.

## 2. Results

### 2.1. Rotenone Treatment Reduced the Neurite Outgrowth and Amount of Dopaminergic Cells

In order to define a rotenone toxicity experimental paradigm, we caused oxidative cellular stress by application of different rotenone concentrations on primary mesencephalic neurons. Cell cultures were treated with rotenone diluted in dimethyl sulfoxide (DMSO) and phosphate buffered saline (PBS) to the final concentration of 10 nM, 20 nM, 40 nM, and 100 nM. DMSO in PBS was added to the control group, but did not exceed 0.01% of DMSO to avoid a toxic effect on the neurons. In order to ensure that the neurons would not impair their regenerative potential due to excessive damage, we titrated the reduction of neurite length to a maximum of 40%.

The total neuron length was reduced by rotenone in a dose-dependent manner. While 10 nM rotenone had no significant effect on the neuritic network compared to the control group (73 ± 24%), 20 nM rotenone led to a relative neurite length reduction to 68 ± 22%. Treatment with 40 nM rotenone significantly decreased the neurite length to 57 ± 16% in the neuritic network, and the treatment with 100 nM rotenone caused a highly significant reduction of the neuron length to 34 ± 17% compared to the control (Figure 1a,b). Besides the effect of different rotenone concentrations on the neurites, we also analyzed the impact on the number of tyrosine hydroxylase (TH) positive dopaminergic cells related to the amount of 4,6-diamidino-2-phenylindole (DAPI) labelled cell nuclei. The concentration of 100 nM rotenone led to a substantial decrease to 31 ± 6% of the control (*p* = 0.058). The effect of the other concentrations was not significant (Figure 1a,c). None of the rotenone concentrations had a significant effect on non-TH cells (Figure 1d).

For the further experiments, we decided to use 20 nM rotenone as lesioning treatment that caused substantial damage with a 32% reduction of the neuritic network.

### 2.2. Fasudil and the SCFA Propionic Acid Have Beneficial Effects on Rotenone Treated Dopaminergic Neurons

Next, we investigated the effect of the ROCK inhibitor Fasudil and the SCFA propionic acid on primary midbrain dopaminergic neurons in six independent experiments. The cells were treated with 20 nM Fasudil, 300 μM propionic acid, or a combination of both on days 1 and 3. Rotenone was added in a concentration of 20 nM to the culture on day 3 for 48 h. The outgrowth of neurites was measured for all conditions, related to the number of cells in the visual field and normalized to the PBS/DMSO treated control group. Additionally, we determined the number of TH+ cells normalized to the control group. To examine the effect on non-TH cells, we counted the DAPI nuclei and subtracted the number of TH+ cells for all conditions and normalized them to the control.

Rotenone 20 nM treatment led to a reduction of the total neurite length of 37 ± 12% compared to the control group, which was set to 100%. ROCK inhibition was able to effectively prevent this damaging effect as Fasudil increased the total neurite length under rotenone treatment. In addition to this growth-promoting effect, Fasudil also significantly increased the percentage of TH+ cells (163 ± 50%) compared to the rotenone treated control group (66 ± 34%). These results indicate a neurite outgrowth promoting effect of the ROCK inhibition on rotenone treated dopaminergic neuron in vitro (Figure 2).

Besides ROCK inhibition with Fasudil, we additionally examined the effect of the SCFA propionic acid on the rotenone treated dopaminergic neurons. The neurites in the cultures treated with 300 µM propionic acid reached 112 ± 41% outgrowth relative to the control group. Compared to the rotenone control group, neurites growth was increased but did not reach significance (*p*-value of 0.105). However, the percentage of surviving TH+ cells normalized to control was significantly higher in the propionic acid treated culture (142 ± 36%) than in the rotenone treated control group (66 ± 34%) (Figure 2). 

The combination of Fasudil and propionic acid had no additional effect on the outgrowth of neurons or the number of TH+ cells. Neither rotenone nor the treatment with propionic acid or Fasudil had a significant effect on the number of non-TH cells (Figure 2d).

### 2.3. Relative Gene Expression for Neuronal and Dopaminergic Proteins under Rotenone Stress

The primers for the rtPCR were chosen to analyze and characterize the function of primary neurons and to evaluate the effect of Fasudil and propionic acid on midbrain cells under rotenone stress. The ΔΔCt method corrected by the efficiency was used to calculate the relative gene-expression, which was normalized to the housekeeping gene beta-actin (βAct) and the control group. Treatment with 20 nM rotenone for 48 h affected the expression of different genes compared to the control group, which was set to 1. The gene for dopamine transporter (DAT) was significantly less expressed in the rotenone treated control group (0.59 ± 0.19). This effect could not be prevented by inhibition of ROCK or by treatment with propionic acid. Rotenone treatment also led to a significantly decreased gene expression of microtubule-associated protein 2 (MAP2) (0.77 ± 0.12). Neither Fasudil nor propionic acid affected this alteration. Furthermore, expression of the gene for synaptophysin (SYP) was decreased by treatment with 20 nM rotenone (0.72 ± 0.13). Interestingly, 20 µM Fasudil could counteract this effect and significantly increased the gene expression to 0.97 ± 0.18. Rotenone treatment had no significant effect on the expression of TH genes but—compared to the control group—propionic acid itself (2.12 ± 0.78) and in combination with Fasudil (2.37 ± 0.82) increased the gene expression under treatment with rotenone. In comparison to the rotenone treated control group (1.24 ± 0.34), propionic acid and its combination with Fasudil increased TH gene expression.

The expression of the aSyn gene only slightly increased under rotenone treatment (1.34 ± 0.31), but in combination with propionic acid its relative gene expression was significantly increased to 1.53 ± 0.30 compared to the control group. The gene expression under treatment with ROCK inhibitor Fasudil and rotenone (1.02 ± 0.18) was significantly less than in the propionic acid treated group. Furthermore, combination of propionic acid and Fasudil under rotenone treatment (0.96 ± 0.21) tended to decrease aSyn gene expression compared to the rotenone control group (*p* = 0.051) (Figure 3).

### 2.4. Western Blot Analyses of Neuroprotective and Regenerative Proteins

Western blot analysis was performed on in vitro day 5 in five independent experiments. Hereby, we sought to examine the effect of rotenone and the treatment with ROCK inhibitor Fasudil and propionic acid on protein levels of important neuroprotective and neuroregenerative pathways including the signal transducer and activator of transcription (STAT) 3, Akt, and TH. The protein expression of STAT3 was increased by rotenone treatment with propionic acid compared to the untreated control. Nevertheless, no significant change in the pSTAT3/STAT3 ratio could be measured in any group. Although we found slight alterations in the levels of protein expression for TH and Akt, these did not reach significant levels of difference including analysis of phosphorylated and the non-phosphorylated proteins (Figure 4).

## 3. Discussion

In our study, we employed a toxin-based in vitro model for PD and evaluated lesioning effects of rotenone on mesencephalic dopaminergic neurons. Our analysis examined the influence of ROCK inhibition and SCFA on neurite outgrowth and on cell survival under rotenone stress. In addition, we examined underlying signaling pathways both on the gene expression and on the protein expression level of mesencephalic neurons.

In the first step, we examined the lesioning effect of different rotenone concentrations on neurite outgrowth of primary dopaminergic cells. Rotenone is known to induce neuron damage and apoptosis via inhibition of the electron transport chain in mitochondria which leads to reduced adenosine triphosphate (ATP) levels and oxidative stress through reactive oxygen species (ROS) [18,26] (Figure 5). We found a dose-dependent reduction of the neuritic network. The final rotenone concentration of 20 nM was used for further experiments because it led to a neurite net reduction of moderate intensity of about 30% without lesioning dopaminergic cells to sublethally. This is in line with other studies that have employed rotenone on primary dopaminergic cell cultures [27,28].

In a design to evaluate neurite outgrowth promoting behavior we applied two different compounds to the rotenone paradigm in vitro. Treatment with ROCK inhibitor Fasudil could not only rescue the damaged neurites but also induce robust neuritic outgrowth, which exceed the mean neurite length in the control culture. Additionally, ROCK inhibition led to an increased survival of dopaminergic neurons. In previous studies, it could be shown that ROCK inhibition is also pro-regenerative in other toxin-based models such as MPTP/MPP+ based in vitro and in vivo experiments [25,29]. We were able to transfer these findings to our rotenone based in vitro experiments. 

The effect of propionic acid on mesencephalic dopaminergic neurons was the second focus in this study. Propionic acid was able to induce neuritic outgrowth under rotenone treatment. Besides this neurite outgrowth promoting effect, propionic acid could significantly increase the number of TH-positive dopaminergic under rotenone treatment (Figure 5). The direct influence of SCFA on neurons of the central nervous system is not sufficiently investigated yet. One study showed proliferative effects of SCFA on human neural progenitor cells in in vitro experiments [30] but further information of other cellular systems is still lacking. An additive or even synergistic effect of propionic acid and Fasudil could not be observed for neurite outgrowth or for preservation of dopaminergic TH+ cells. Cellular neuroprotection with an increase in dopaminergic cell survival under rotenone stress was found to be significant for both propionic acid and Fasudil. However, a combinatorially augmented cellular protection was not found possibly because cellular protection was already maximized under each single treatment. 

Analyzing gene expression profiles in our experimental setup we found that rotenone had a decreasing effect on the gene expression of DAT, MAP2 and SYP. Other studies showed that, DAT protein expression was downregulated in in vivo experiments with rotenone in rats [31]. Rotenone seems to have an influence on the regulation of gene and protein expression of dopamine re-uptake. MAP2 as a late neuronal marker stabilizes microtubules. Rotenone has been shown to have inhibiting effects on microtubule assembly in in vitro experiments [32]. Furthermore, downregulation of the protein leads to activation of apoptotic pathways and was shown in in vivo studies of rotenone effects in rats [33]. Decrease of SYP expression has already been described in in vivo experiments with rotenone treated rats [34]. While Fasudil had no beneficial influence on DAT, MAP2 or TH gene expression under rotenone stress, it significantly reduced the effect of rotenone on SYP and therefore implies a protective effect on the synaptic integrity [35]. Gene expression of TH was not reduced by rotenone but the treatment with propionic acid alone and in combination with Fasudil tended to increase the TH-RNA transcription. SCFA seem to affect the neurotransmitter synthesis in dopaminergic neurons through regulation of tryptophan 5-hydroxylase 1 and increase TH levels [22,36]. Like Fasudil propionic acid had no effect on the relative gene expression of DAT. Interestingly, rotenone treatment activated gene expression of aSyn. In this context, the combination of rotenone and propionic acid led to a significant difference in comparison to the control group. The potential of SCFA to induce aSyn aggregation in vivo [10] and to increase expression of endogenous aSyn in neurons [37] has been previously described. In this study propionic acid had an activating effect on transcriptional mechanisms for aSyn expression. The physiological function of endogenous aSyn is not completely understood but it is suggested to regulate presynaptic vesicles [38,39] and to influence vesicular dynamics in cells [40,41]. Therefore, the influence of SCFA on aSyn levels, protein aggregation, and associated effects on dopaminergic neuron physiology have to be analyzed in detail in subsequent studies. Concerning Fasudil, an attenuating effect on aSyn aggregation was already described [42]. Our findings thus show a regulating effect of aSyn gene expression after Fasudil and propionic acid treatment under rotenone stress (Figure 5).

The analysis of neuroprotective and neuroregenerative protein expression pathways resulted in no robust modifications of pathways such as the Akt cell survival or the STAT3 signaling pathway in this paradigm. A tendency to an increase of the pAkt/Akt ratio under treatment with rotenone and Fasudil was found. Regarding ROCK inhibitor treatment other publications describe an increase in Akt activation via phosphorylation under MPP+ treatment [25]. The application of propionic acid under rotenone stress led to a significant increase of STAT3 protein expression, but had no influence on the ratio of pSTAT3/STAT3 nor on the pAkt/Akt ratio. Thus, other not yet identified protective mechanisms seem to activate by propionic acid, which, however, do not add to the Fasudil protective effects.

A special characteristic of the study is, that dopaminergic cells represent only a small proportion of cells in the mixed mesenchymal culture paradigm of 2–5% on average. However, cellular homeostasis strongly depends on the presence of non-dopaminergic cells, too. Therefore, analyses for gene and protein expression do reflect the function of all cell types in the culture system. In following studies, these analyzes would be of interest in separated subtypes of cells after cell sorting or even at the single cell level at different time points.

The therapeutic potential of SCFA in neurological diseases is subject of interest in many studies. Due to its inhibiting effects on histone deacetylase (HDAC), which lead to inhibition of gene transcription, SCFA show beneficial effects in neurodegenerative diseases such as Alzheimer’s disease [43,44]. Furthermore, SCFA are already used in other neurological diseases like valproic acid is used as anti-epileptic drug since 1967 [45]. Besides its promoting effect on neural progenitor cells [30], the direct impact on SCFA on neurons is poorly investigated, but an influence of neurotransmitter synthesis and levels of neurotrophic factors has been observed [22]. Furthermore, post-stroke recovery in a mice model was improved by SCFA treatment through increased spine and synapse densities and improved connectivity but also immunological mechanisms like microglia activation [46]. Studies concerning beneficial effects of propionic acid in particular have primarily focused on its anti-inflammatory properties. In a recent application study in human multiple sclerosis patients, propionic acid substantially increased regulatory T cells and decreased pro-inflammatory Th1 and Th17 cells in serum and feces which was associated with a more beneficial course of disease [47].

In PD research, the role of the immune system has only emerged in recent years. Several preclinical and clinical studies underline the impact of immunological dysregulations like inflammatory cell activation on PD [48]. For example, increased pro-inflammatory Th17 cells can be observed in PD mouse models and human blood and port-mortem brains of PD patients [49,50]. SCFA as has been demonstrated to influence inflammatory cells like microglia or leukocytes and effect the production of pro- and anti-inflammatory cytokines like interleukin 10 (IL-10) or tumor necrosis factor alpha (TNFα) [47,51]. Therefore, the immune system of the PD patients is an important target for further therapeutic research [48]. In a recent analysis of the plasma concentration of propionic acid in PD patients, low propionic acid levels correlated with worse motor function as measured with the Unified Parkinson’s Disease Rating Score part III (UPDRS-III) [52]. Besides studies describing protective and anti-inflammatory influence, other data also demonstrate negative effects of SCFA. In in vivo models of PD, an increase in microglia activation was observed after enhancement of aSyn-mediated pathology by SCFA [10]. Propionic acid has been shown to induce gliosis and neuroinflammation as well as produce changes in behavior like hyperactivity and social behavior impairments in models for neurodevelopmental disorders like autism spectrum disorder in particular [53,54,55]. Therefore, beneficial of harmful effects of SCFA including propionic acid for CNS disease depend on the disease-specific context and have to be explored in more detail especially in neurodegenerative diseases.

In summary, our data on ROCK inhibition and propionic acid treatment in the in vitro rotenone paradigm underline their beneficial potential on dopaminergic neurons. To accurately determine to what extent the effects of both treatments can be neuroprotective or have neuroregenerative potential further analyses have to be added. Our findings accentuate the need for further research focusing on direct influences of enteric metabolites on neuronal cells in the central nervous system. In order to examine, if the neuroprotective potential of Fasudil and SCFA and their combination can be transferred to in vivo experiments, analyses in animal models should be subject of subsequent studies. Furthermore, the recently demonstrated important role of propionic acid in neuroinflammatory diseases such as multiple sclerosis demand a more in-depth analysis of the impact on inflammatory mechanisms in neurodegenerative conditions [47].

## 4. Materials and Methods

### 4.1. In Vitro Assay of Primary Midbrain Dopaminergic Neurons

For the preparation of primary midbrain dopaminergic neurons pregnant Sprague Dawley rats (Charles River Laboratories, Wilmington, MA, USA) were sacrificed on embryonic day 14. Briefly, embryos were removed, midbrains were dissected and the neurons were seeded on poly-D-lysine-coated (Sigma-Aldrich, St. Louis, MO, USA) cover slips and cultured in neurobasal (Life Technologies, Carlsbad, CA, USA) with 1% B27 supplement (Life Technologies), 1% glutamine 200 mM (Thermo Fisher Scientific, Waltham, MA, USA), 1% penicillin/streptomycin (Thermo Fisher Scientific) and 0.1% l-ascorbic acid 200 mM (Sigma-Aldrich) for five days. For the rotenone paradigm, half of the medium was changed on in vitro day 2. On day 3 cells were treated with rotenone with a final concentration of 10 nM, 20 nM, 40 nM, or 100 nM for 48 h to establish a concentration for sufficient cellular degeneration. Rotenone was diluted in 99.9% DMSO (Thermo Fisher Scientific) (in a stock solution of 100 μM. The working solution was further diluted with neurobasal medium. The control was always treated with an equal concentration of DMSO diluted in PBS (Life Technologies), which did not exceed 0.01%. For further experiments, cells were seeded as previously described. On in vitro day 1 and 3, cells were treated with 20 μM Fasudil (Biozol, Eching, Germany), with 300 μM propionic acid (Sigma-Aldrich) or with a combination of both. On day 3, half of the medium was changed and 20 nM rotenone or PBS/DMSO was added to the cultures.

### 4.2. Immunocytochemistry

On in vitro day 5, cells were fixed with 4% of paraformaldehyde for 10 min at room temperature, washed three times with PBS, permeabilized with 0.1% triton (Sigma-Aldrich) in PBS for 5 min and were then blocked with 10% normal goat serum (Biozol) for 10 min at room temperature. 

To visualize dopaminergic neurons, cells were incubated over night at 4 °C with the primary antibody anti-tyrosine hydroxylase (1:1000, Merck Millipore, Burlington, MA, USA). Washing of the cells three times with PBS was followed by incubation with the secondary antibody 488 goat anti-rabbit (1:500, Life Technologies) for 45 min at 37 °C. After two more washes the coverslips were then mounted in DAPI Fluoromount (Biozol) to stain the cell nuclei with DAPI and afterwards analyzed with a fluorescent microscope (Olympus BX51, Hamburg, Germany). In total, six randomized visual fields were analyzed with an x40 objective for each coverslip under blinded conditions and for the morphological analysis the cumulative lengths of the neurites were measured with the plugin NeuronJ of the software ImageJ. In each field the number of TH+ and DAPI positive cells were counted, to calculate a quotient of neurite length per cell and to compare the number of dopaminergic cells. The mean of the six fields was formed and compared.

### 4.3. Real-Time Quantitative PCR

RNA was isolated on day 5 in vitro with the ReliaPrep-Kit (Promega, Madison, WI, USA). After determination of the amount and purity of the RNA with the NanoDrop spectrophotometer ND-100 (Peqlab Biotechnologie GmbH, Erlangen, Germany) 0.5 μg RNA was transcribed into complementary DNA (cDNA) using qScript cDNA SuperMix (Quantabio, Beverly, MA, USA). Microsynth (Balgach, Switzerland) was used to design all primers. Primer sequences are listed in Table 1. The efficiencies of all primers, which were taken as sufficient with a value of 2 ± 0.15, were detected with serial dilution. For relative quantification, the cDNA was diluted 1:10. The experiments were performed with a QuantStudio 3 (Applied Biosystem by Thermo Fisher) with GoTaq qPCR Master Mix containing BRYT Green dye (Promega). Following conditions were used: Polymerase activation step at 95 °C for 2 min, followed by 45 cycles of 2-step qPCR (10 s denaturation at 95 °C and 45 sec annealing and extension). Using the ΔΔCt method corrected by the efficiency the relative gene-expression was calculated by normalizing to the housekeeping gene β-actin and the untreated control.

### 4.4. Western Blot

For the evaluation of intracellular pathways, primary mesencephalic cells were treated as described above. On day 5 the cells were lysed in radioimmunoprecipitation assay buffer (RIPA buffer) containing 25 mM TRIS buffer (pH 8.0) (Trizma base, Sigma-Aldrich), 150 nm sodium chloride (Thermo Fisher Scientific), 1% triton X-100 (Sigma-Aldrich), 1% deoxycholate (Merck Millipore), 0.1% Sodium dodecyl sulfate (SDS) (Biomol, Hamburg, Germany), 1 mM ethylenediaminetetraacetic acid (EDTA) (Titriplex III, Merck Millipore), protease inhibitor (cOmplete™ Protease Inhibitor Cocktail, Sigma-Aldrich) and Phosphatase Inhibitor (PhosStop, Roche, Basel, Switzerland). The Pierce bicinchoninic acid (BCA) protein assay (Thermo Fisher Scientific) was performed to determine the content of protein in each cell sample. SDS polyacrylamide gel electrophoresis (SDS-Pages) was loaded with a constant amount of protein (10 μg) of each condition. The separated proteins were transferred to a nitrocellulose membrane and blocked with 3% milk powder in Tris-buffered saline (TBS) for 1 h at room temperature. Afterwards, membranes were incubated with primary antibodies (anti-phosphorylated STAT3, anti-STAT3, anti-phosphorylated Akt, anti-Akt, all 1:1000; all Cell Signaling Technology, Danvers, MA, USA; anti-TH, 1:1000, Merck Millipore, anti-β-actin, 1:5000, Linaris GmbH, Dossenheim, Germany) over night at 4 °C in TBS/Tween (TBS-T) with milk powder. After three washing steps with TBS-T the membranes were incubated with a secondary peroxidase conjugated antibody (1:10,000) (Sigma-Aldrich). Immobilon Western Chemiluminescent HRP substrate (Merck Millipore) was applied and the chemiluminescence was visualized by G:BOX Chemi XRQ (Syngene, Cambridge, UK). Image Studio Lite 5.0 (LI-COR Biosciences, Lincoln, NE, USA) was used to measure the intensity.

### 4.5. Statistical Analysis

The data are represented as mean ± SD or SEM. For statistical analyses, Prism 8 (GraphPad Software, San Diego, CA, USA) was used. Normal distribution was checked with the Kolmogorov–Smirnov test and homoscedasticity was tested with Brown-Forsythe and Bartlett’s test. When values are normally distributed, one-way ANOVA was performed with Tukey’s post hoc test. Otherwise, Kruskal–Wallis test with Dunn’s correction for multiple comparisons was used. Significances are indicated with * *p* < 0.05, ** *p* < 0.01, *** *p* < 0.001.

## Figures and Tables

**Figure 1 molecules-25-02502-f001:**
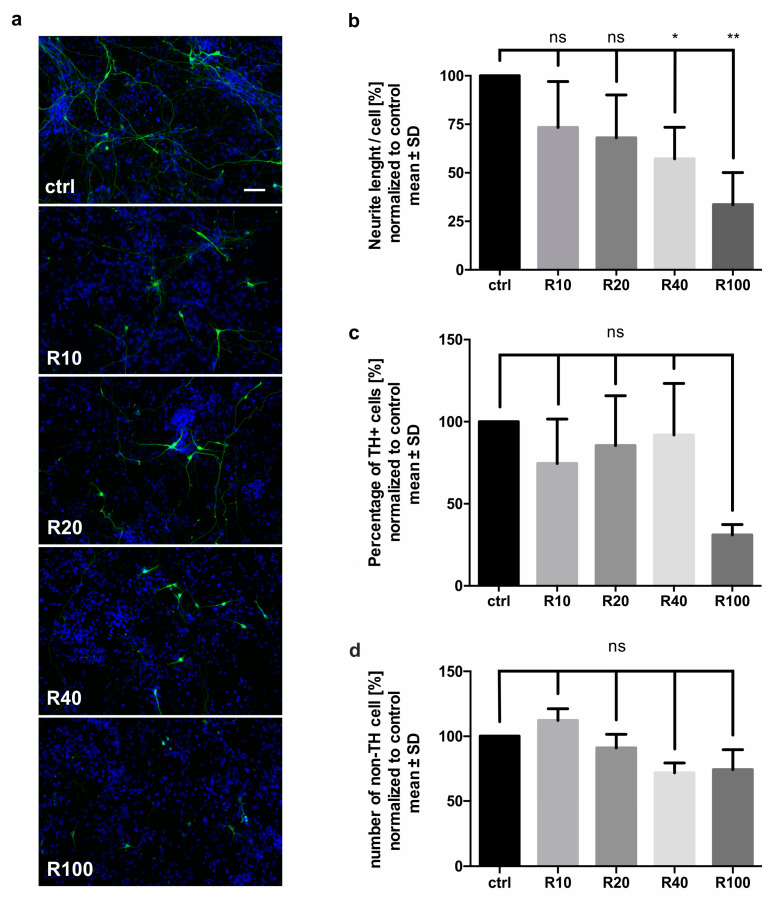
Dose-dependent effect of different rotenone concentrations (10 nM, 20 nM, 40 nM, and 100 nM) on primary mesencephalic neurons: (**a**) Representative micrographs of dopaminergic midbrain neuron cultures 48 h after the treatment with rotenone labeled against tyrosine hydroxylase (TH) (green). Cell nuclei were stained with DAPI (blue); Scale bar = 50 µm. (**b**) Rotenone lesioning effect on the neuritic network: Mean neurite length per cell was measured in six visual fields and normalized to control (ctrl). (**c**) Percentage of TH-positive cells in rotenone treated cultures normalized to control. (**d**) Number of non-TH cells in cultures treated with different rotenone concentrations normalized to control. Bars represent mean ± SD. ns: not significant. * *p* < 0.05, ** *p* < 0.01. R10 = 10 nM, R20 = 20 nM, R40 = 40 nM and R100 = 100 nM rotenone respectively.

**Figure 2 molecules-25-02502-f002:**
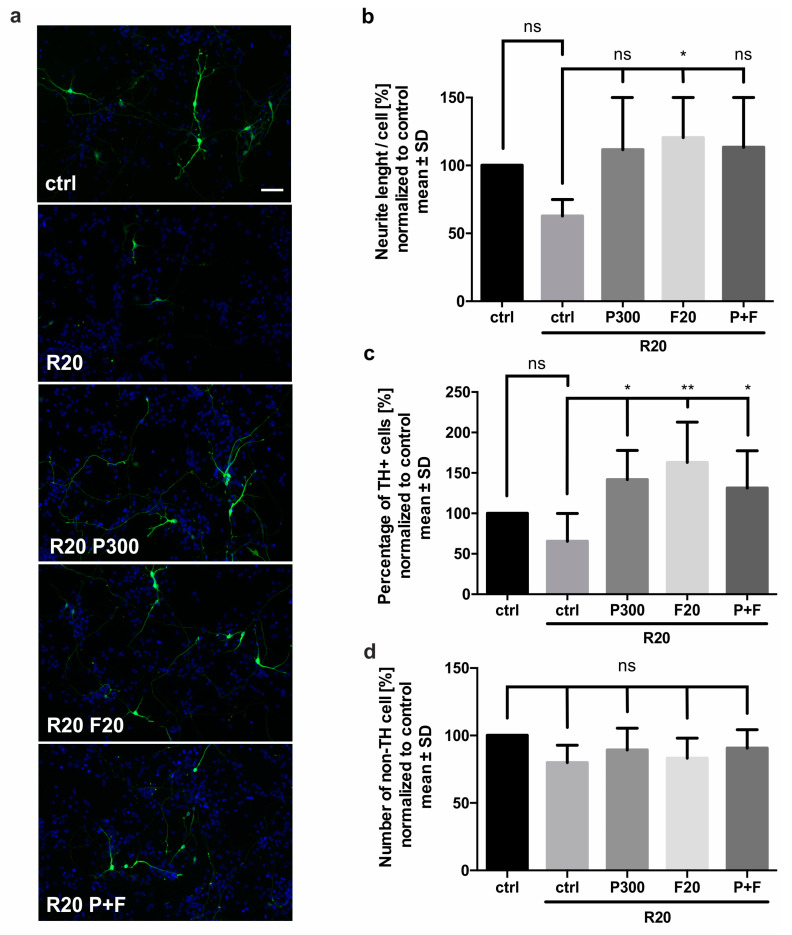
Effect of Rho-associated protein kinase (ROCK) inhibition and short chain fatty acid (SCFA) on rotenone treated dopaminergic neurons in six independent experiments. (**a**) Representative micrographs of TH+ dopaminergic neurons (green) and unspecific cell nuclei (blue) under different conditions. Cultures were treated with 20 nM rotenone (R20) and 300 µM propionic acid (P300) or 20 µM Fasudil (F20) or the combination (P+F); Scale bar = 50 µm. (**b**) Quantification of the neuritic outgrowth of dopaminergic neurons. Measured in six randomized visual fields per condition normalized to the control (ctrl). (**c**) Percentage of TH+ cells under the described conditions. (**d**) Number of non-TH cells normalized to the ctrl. Bars represent mean ± SD. TH = tyrosine hydroxylase. ns = not significant. * *p* < 0.05, ** *p* < 0.01.

**Figure 3 molecules-25-02502-f003:**
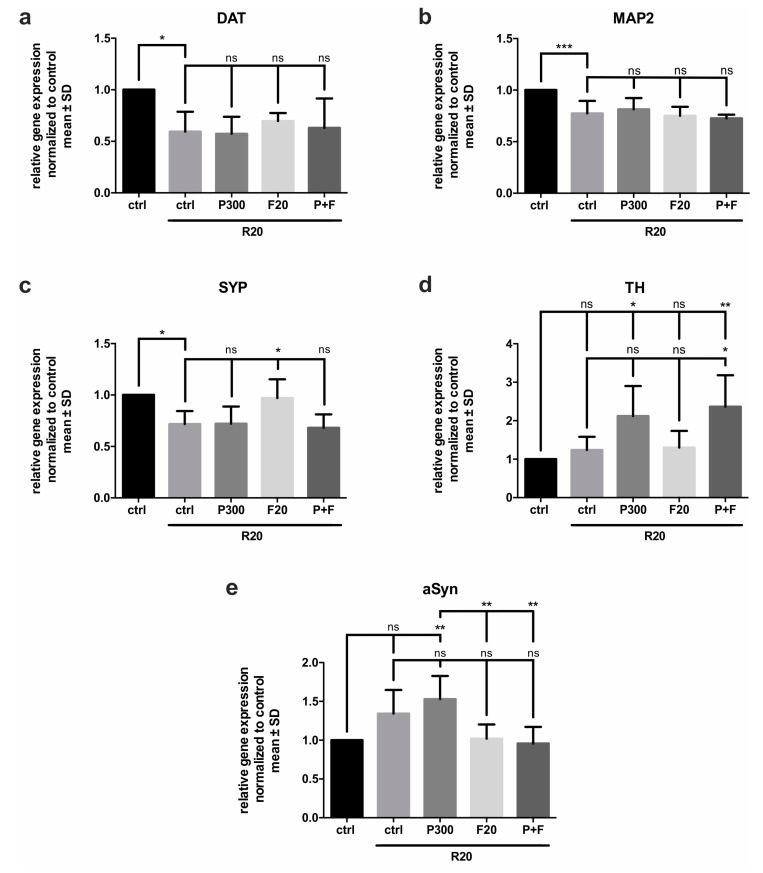
Relative gene expression after treatment with propionic acid (P300) and Fasudil (F20) under 48 h rotenone (R20) exposure in six independent experiments. (**a**–**f**) Quantification of gene expression compared to housekeeping gene β-actin and normalized to the control (ctrl). Presented are (**a**) dopamine transporter (DAT), (**b**) microtubule associated protein (MAP) 2, (**c**) synaptophysin (SYP), (**d**) tyrosine hydroxylase (TH), and (**e**) alpha Synuclein (aSyn). Bars represent mean ± SD. ns = not significant. * *p* < 0.05, ** *p* < 0.01, *** *p* < 0.01.

**Figure 4 molecules-25-02502-f004:**
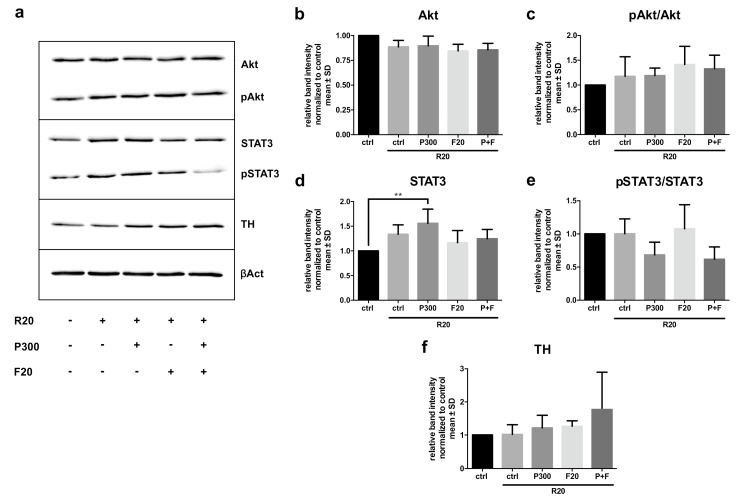
Activation of neuroprotective and neuroregenerative signaling pathways in midbrain dopaminergic neurons treated with or without 20 nM rotenone (R20) and 300 µM propionic acid (P300) and 20 µM Fasudil (F20). (**a**) Representative immunoblots showing the regulation of Akt, pAkt, signal transducer, and activator of transcription (STAT3), pSTAT3, tyrosine hydroxylase (TH), and β-actin (βAct). 10 µg protein per condition were used. (**b**–**f**) Intensity of immunoblot bands from five independent experiments for (**b**) Akt, (**c**) pAkt/Akt ratio, (**d**) STAT3, (**e**) pSTAT3/STAT3 ratio, and (**f**) TH compared to βAct and normalized to the control (ctrl). Bars represent mean ± SD. ** *p* < 0.01.

**Figure 5 molecules-25-02502-f005:**
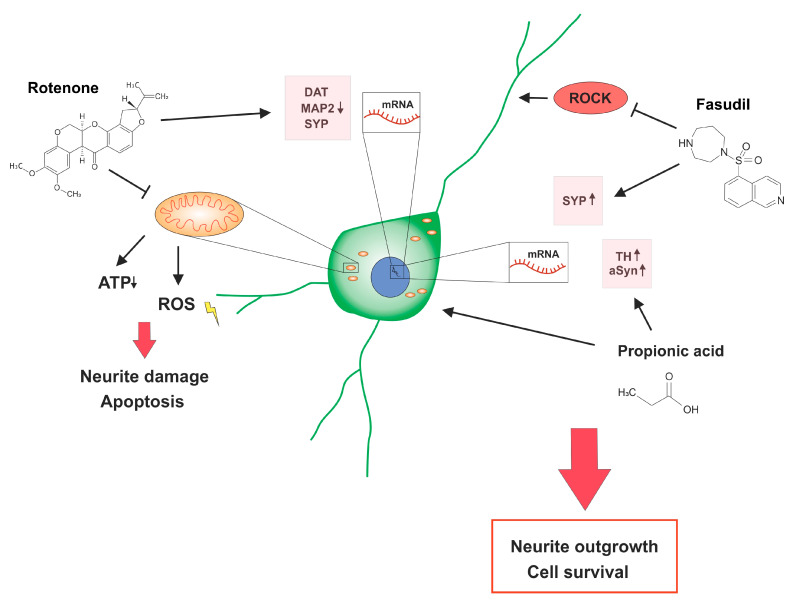
Rotenone toxicity and neuroprotective effects of propionic acid and Fasudil on dopaminergic neurons. Rotenone inhibits the mitochondrial respiratory chain which leads to decreased adenosine triphosphate (ATP) levels, oxidative stress through reactive oxygen species (ROS) and ultimately to neurite damage and apoptosis. Gene expression levels of the dopamine transporter (DAT), microtubule associated protein 2 (MAP2), and synaptophysin (SYP) were reduced under rotenone treatment. Propionic acid increased tyrosine hydroxylase (TH) and alpha Synuclein (aSyn) gene expressions, while treatment with the Rho Kinase (ROCK) inhibitor Fasudil led to increased SYP gene expression. Both treatments had positive effects on neurite outgrowth and cell survival under rotenone toxicity.

**Table 1 molecules-25-02502-t001:** Primer sequences used for rtPCR.

Primer	Sequence
βAct fw	5′-GAAGTGTGACGTTGACATCCG-3′
βAct rev	5′-TCCACACAGAGTACTTGCGC-3′
TH fw	5′-CCTTTGACCCAGACACAGCA-3′
TH rev	5′-TCAATGGCCAGTGTGTACGG-3′
DAT fw	5′-GCCTATGCCATCACACCTGA-3′
DAT rev	5′-CGTGGGTTTCCTTTGCGATC-3′
aSyn fw	5′-CTGTGGACCCTAGCAGTGAG-3′
aSyn rev	5′-AGCACTTGTACGCCATGGAA-3′
SYP fw	5′-CACCTCCTTCTCCAATCAGATGT-3′
SYP rev	5′-GGGTGAATGTAGGGCTCAGAC-3′
MAP2 fw	5′-TGGACGCGTGAAGATTGAGA-3′
MAP2 rev	5′-TGCATGCTCTCGGAAGTTCA-3′

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
