# Peer review of "Propionic Acid and Fasudil as Treatment against Rotenone Toxicity in an In Vitro Model of Parkinson’s Disease"

_molecules, 2020, doi:10.3390/molecules25112502_

Round 1

Reviewer 1 Report

This pioneer study provides interesting data on potential novel therapeutic perspectives in Parkinson’s disease (PD). Recently, there has been a growing understanding of the role of gut microbiota alterations and microbiota-derived short-chain fatty acids (SCFAs) in the pathogenesis of PD. Therefore, the evaluation of the influence of propionic acid (PPA) on dopaminergic neurons in an in vitro model of PD is particularly interesting, although quite controversial taking into account some inconsistent findings in that field. The authors employed rotenone-based in vitro model of PD to investigate neuroprotective effects of Fasudil (Rho kinase inhibitor) and PPA on primary neurons in cell morphological assays, cell survival as well as gene and protein expression. In the first part of the experiments the authors confirmed that rotenone reduced the total neuron length in a dose-depended manner. Both Fasudil and PPA enhanced neurite outgrowth and increased cell survival of dopaminergic neurons. Moreover, Fasudil augmented gene expression of synaptophysin, while PPA substantially increased gene expression of tyrosine hydroxylase. The results have provided new insight into neuroprotective and neuroregenerative pathways involved in that PD model.
The paper is well written, with clearly defined objectives. The large dataset is presented in an approachable manner with adequate visualizations summarizing the results. However, there are some concerns listed below that are related mainly to the study limitations not properly highlighted in the discussion. They should be addressed in the revised version of the paper before it is published.

Specific comments:

  1. In the introduction it is stated with respect to SCFAs (line 46) that “heterogeneous descriptive results” have been reported. However, not enough attention is paid to the potential detrimental effect of SCFAs, and in particular PPA. It should be more extensively discussed, since there are numerous studies confirming the pathogenic role of SCFAs, and specifically PPA in neuroinflammation. For example SCFAs may exacerbate α-synuclein pathology and microglial activation in a mouse model of PD as shown recently by Mazmanian’s group [Sampson et al., Cell 2016]. PPA has been reported to induce gliosis and neuroinflammation in autism spectrum disorder [Abdelli et al., Sci Reports 2019]. It has also been shown that PPA administration induced abnormal neural cell organization, which may lead to autism-like neurobehavioral changes in a rat model [Choi et al., PloS One 2018]. Moreover, intraventricular PPA administration in an experimental model can induce hyperkinetic movements [MacFabe et al., Bev Brain Res 2007].
  2. Please explain how the dose of PPA (300 µM) applied in the current study was established and why only the one dose was used.
  3. A key question is if the PPA neuroprotective effect is dose-dependent. It has not been investigated in the current study that constitutes another substantial limitation. Using different doses of PPA would allow to verify its beneficial vs detrimental influence on dopaminergic neurons. This is a critically important issue in light of many data indicating its neurotoxic potential as mentioned above.
  4. One of the most intriguing findings reported in the study is the observation that the expression of the αSyn gene only slightly increased under rotenone treatment, but in combination with PPA its relative gene expression was significantly increased compared to the control group (Fig 3e). This finding is not consistent with the neuroprotective effect of PPA and should be clearly pointed out and more carefully discussed. In the current version of the paper no clear interpretation of this finding is given.
  5. Although the authors stated in the conclusion that “subsequent in vivo analyses in animal models are justified”, it should be more clearly emphasized that the results of in vitro study cannot be directly extrapolated to in vivo observations and therefore the conclusions should be drawn with caution.

Minor comments:

  1. Line 51 – Instead of “cultured mesenchymal neurons” probably it should be “cultured mesencephalic neurons” or “neurons in mesenchymal culture” – please verify
  2. In Figure 1c for rotenone concertation of 100 nM (i.e. R100) a significance mark (*) corresponding to p-value is missing. As stated in lines 98-99: ”The concentration of 100 nM rotenone led to a substantial decrease to 31.08 ± 6.24% of the control.” By the way, the given accuracy of percentage change is inapt.

Author Response

Dear Reviewer 1,

We thank you for your insightful comments. We answered them in the following point-to-point reply.

Specific comments:

1. In the introduction it is stated with respect to SCFAs (line 46) that “heterogeneous descriptive results” have been reported. However, not enough attention is paid to the potential detrimental effect of SCFAs, and in particular PPA. It should be more extensively discussed, since there are numerous studies confirming the pathogenic role of SCFAs, and specifically PPA in neuroinflammation. For example SCFAs may exacerbate α-synuclein pathology and microglial activation in a mouse model of PD as shown recently by Mazmanian’s group [Sampson et al., Cell 2016]. PPA has been reported to induce gliosis and neuroinflammation in autism spectrum disorder [Abdelli et al., Sci Reports 2019]. It has also been shown that PPA administration induced abnormal neural cell organization, which may lead to autism-like neurobehavioral changes in a rat model [Choi et al., PloS One 2018]. Moreover, intraventricular PPA administration in an experimental model can induce hyperkinetic movements [MacFabe et al., Bev Brain Res 2007].

Answer: We thank the reviewer for this insightful comment. We agree, that the role and effect of SCFA are controversially discussed and effects depend on disease-specific differences. We therefore, added a new paragraph comprising the most recent data on the topic to the discussion (lines 349-356). 

2. Please explain how the dose of PPA (300 µM) applied in the current study was established and why only the one dose was used.

Answer: The dose of propionic acid in our pilot study has been selected based on results of a previous study of our co-working research group which applied this dosage effectively inin vitro T cell differentiating assays, on iPSC-derived motoneuron cells and on an in vivo EAE model of multiple sclerosis (Haghikia et al. Dietary Fatty Acids Directly Impact Central Nervous System Autoimmunity via the Small Intestine. Immunity.2015).

3. A key question is if the PPA neuroprotective effect is dose-dependent. It has not been investigated in the current study that constitutes another substantial limitation. Using different doses of PPA would allow to verify its beneficial vs detrimental influence on dopaminergic neurons. This is a critically important issue in light of many data indicating its neurotoxic potential as mentioned above. 

Answer: This is an important comment. We agree with the reviewer, that several dose ranges of propionic acid have to be closely examined, as there can be differential effects in the model systems. As mentioned above we applied the propionic acid concentration after dose optimization in two in vitro and in analogy to an EAE in vivo model of multiple sclerosis (Haghikia et al. Immunity 2015). To analyze different concentrations of SCFA, new experiments in our in vitro model are currently planned.

4. One of the most intriguing findings reported in the study is the observation that the expression of the αSyn gene only slightly increased under rotenone treatment, but in combination with PPA its relative gene expression was significantly increased compared to the control group (Fig 3e). This finding is not consistent with the neuroprotective effect of PPA and should be clearly pointed out and more carefully discussed. In the current version of the paper no clear interpretation of this finding is given.

Answer: We thank the reviewer for this comment. In our study propionic acid led to increased aSyn gene transcription. We agree with the author that misfolded and aggregated aSyn is a hallmark of PD and is known to have neurotoxic effects on dopaminergic neurons. The cause for aSyn aggregation and the physiological function of endogenous aSyn are still poorly understood (Killinger et. al., Nature 2019). Therefore, at this point a sharp interpretation of our findings regarding increased aSyn gene expression is difficult. In subsequent studies, aSyn aggregation analyses have to be integrated. To point out our findings we added a novel paragraph to the discussion (lines 274-292).

5. Although the authors stated in the conclusion that “subsequent in vivo analyses in animal models are justified”, it should be more clearly emphasized that the results of in vitro study cannot be directly extrapolated to in vivo observations and therefore the conclusions should be drawn with caution.

Answer: We thank the author for this comment. We rephrased the sentence in our conclusions to: “In order to examine, if the neuroprotective potential of Fasudil and SCFA and their combination can be transferred to in vivo experiments, analyses in animal models should be subject of subsequent studies” (lines 362-364).

Minor comments:

1. Line 51 – Instead of “cultured mesenchymal neurons” probably it should be “cultured mesencephalic neurons” or “neurons in mesenchymal culture” – please verify

Answer: We thank the reviewer for the comment. The term was changed to “cultured mesencephalic neurons” (line 51).

2. In Figure 1c for rotenone concertation of 100 nM (i.e. R100) a significance mark (*) corresponding to p-value is missing. As stated in lines 98-99: ”The concentration of 100 nM rotenone led to a substantial decrease to 31.08 ± 6.24% of the control.” By the way, the given accuracy of percentage change is inapt.

Answer: We thank the reviewer for this comment. The p-value for the concentration of 100 nM in comparison to the control is p = 0.0582. Therefore, the difference is not significant. To point this out, we added the p-value to the results in line 106. We furthermore, adjusted the accuracy of percentage through reduction of the decimal places.

Reviewer 2 Report

The author has submitted a research manuscript of effects of a Rho kinase inhibitor fasudil and propionic acid on rotenone-lesioned cultured primary mesencephalic cells prepared from SD rats, and found that both propionic acid and fasudil rescued the rotenone-induced cell damage to some extent by measured in terms of some biomarkers of dopaminergic cells such as tyrosine hydroxylase. Interestingly, the effects of propionic acid on cell lesions were not similar to those of fasudil. This issue is of interest, and impact of their research is considered to be strong. My overall concern with the research manuscript describing a novel, possible pharmacotherapy for Parkinson’s disease is that information provided may offer something substantial that helps advance our understanding of medicines targeting on dopaminergic cell degradation in Parkinson’s disease.

I have a couple of following comments to improve this manuscript for consideration:

Firstly, the author emphasized an importance of both propionic acid and fasudil for treating the rotenone-lesioned cell damage, but these two agents had no additive effects on protecting cell damage. Please explain this evidence by putative molecular mechanisms so that this research would better be enhanced its strength. Next, subtitles of this manuscript should be illustrated to clarify the results: for instance, the authors would present the figure illustrating that how rotenone influences to dopaminergic cells, and then how both propionic acid and fasudil rescued the rotenone-induced cell damage. The information will help readers to understand their results easier than the present form.

Author Response

Dear Reviewer 2,

we thank you for your important comments. We answered them in the following point-to-point reply.

  1. Firstly, the author emphasized an importance of both propionic acid and fasudil for treating the rotenone-lesioned cell damage, but these two agents had no additive effects on protecting cell damage. Please explain this evidence by putative molecular mechanisms so that this research would better be enhanced its strength.

Answer: We thank the reviewer for this important and insightful comment. In our study propionic acid and Fasudil had neuroprotective effects, but no additive effects could be observed. There can be various reasons for this finding. There could be a “saturation” of pro-regenerative and protective effects of the dopaminergic neurons. Both treatments, led to increased neuritic outgrowth and higher percentage of TH+ cell even compared to the control group (mean > 100 %). Possibly, the growth promoting and neuroprotective potential is already exploited under treatment with the Fasudil or propionic acid and therefore, a combination of both had no additive effect. We expanded two paragraphs about this topic in the discussion (lines 251 – 256 and lines 299 – 301).

  1. Next, subtitles of this manuscript should be illustrated to clarify the results: for instance, the authors would present the figure illustrating that how rotenone influences to dopaminergic cells, and then how both propionic acid and fasudil rescued the rotenone-induced cell damage. The information will help readers to understand their results easier than the present form.

Answer: We thank the reviewer for this helpful comment. To illustrate our results and make our findings easier to understand, we added a novel figure to the manuscript (figure 5, line 308).

Round 2

Reviewer 2 Report

The authors have properly addressed all the issues raised in reviewers' comments including me. The revised manuscript is now suggested to be accepted for publication in Molecules.